# The Enrichment of Specific Hair Follicle-Associated Cell Populations in Plucked Hairs Offers an Opportunity to Study Gene Expression Underlying Hair Traits

**DOI:** 10.3390/ijms24010561

**Published:** 2022-12-29

**Authors:** Rakan Naboulsi, Jakub Cieślak, Denis Headon, Ahmad Jouni, Juan J. Negro, Göran Andersson, Gabriella Lindgren

**Affiliations:** 1Department of Animal Breeding and Genetics, Swedish University of Agricultural Sciences, 750 07 Uppsala, Sweden; 2Department of Genetics and Animal Breeding, Poznan University of Life Sciences, 60-637 Poznań, Poland; 3The Roslin Institute and Royal (Dick) School of Veterinary Studies, University of Edinburgh, Edinburgh EH25 9RG, UK; 4Department of Evolutionary Ecology, Doñana Biological Station, CSIC, 41092 Seville, Spain; 5Center for Animal Breeding and Genetics, Department of Biosystems, KU Leuven, 3001 Leuven, Belgium

**Keywords:** equine, skin, hair, follicle, RNA-seq

## Abstract

Gene expression differences can assist in characterizing important underlying genetic mechanisms between different phenotypic traits. However, when population-dense tissues are studied, the signals from scarce populations are diluted. Therefore, appropriately choosing a sample collection method that enriches a particular type of effector cells might yield more specific results. To address this issue, we performed a polyA-selected RNA-seq experiment of domestic horse (*Equus ferus caballus*) plucked-hair samples and skin biopsies. Then, we layered the horse gene abundance results against cell type-specific marker genes generated from a scRNA-seq supported with spatial mapping of laboratory mouse (*Mus musculus*) skin to identify the captured populations. The hair-plucking and skin-biopsy sample-collection methods yielded comparable quality and quantity of RNA-seq results. Keratin-related genes, such as KRT84 and KRT75, were among the genes that showed higher abundance in plucked hairs, while genes involved in cellular processes and enzymatic activities, such as MGST1, had higher abundance in skin biopsies. We found an enrichment of hair-follicle keratinocytes in plucked hairs, but detected an enrichment of other populations, including epidermis keratinocytes, in skin biopsies. In mammalian models, biopsies are often the method of choice for a plethora of gene expression studies and to our knowledge, this is a novel study that compares the cell-type enrichment between the non-invasive hair-plucking and the invasive skin-biopsy sample-collection methods. Here, we show that the non-invasive and ethically uncontroversial plucked-hair method is recommended depending on the research question. In conclusion, our study will allow downstream -omics approaches to better understand integumentary conditions in both health and disease in horses as well as other mammals.

## 1. Introduction

Skin is a highly complex tissue that can be affected by numerous genetic diseases [1,2]. It consists of many different cell types and structures, such as the hair follicles (HF), that contain and surround hair roots. HFs serve as a reservoir of stem cells and can be considered a dynamic mini-organ. This complexity of the skin tissue is one of the reasons why transcriptome studies that aim to understand how healthy and pathological skin cells behave, as well as during the development of an organism, are usually challenging. However, certain studies, such as pigmentation and mane hair color investigations, as well as disorders affecting the hair shaft, might be facilitated and more appropriately performed by narrowing tissue sampling on the HF. The non-invasive sample-collection method of hair plucking would be one possible and easy way to zoom in on the fascinating structures of HFs and to use plucked hairs as the tissue source for gene expression analysis.

Human plucked HFs have previously been used as a source for high-quality total-RNA extracts suitable for RNA-seq experiments [3]. Yet, this study did not investigate the link between gene expression and the various cell populations in the skin. However, single-cell RNA-seq experiments combined with spatial transcriptomics from the same tissue, may identify this link. However, such an experiment might not be needed especially if the punch biopsy samples can be substituted by the easy-to-obtain plucked hair samples. For this reason, plucked hair samples seem to be especially relevant for studies focusing on HF-related phenotypes such as hair growth, pigmentation, structure, and composition of the hair shaft.

We hypothesize that zooming in, achieved by hair plucking, on a specific anatomical structure of the skin—the HF—leads to an enrichment in the HF-associated cell populations and thus a reduced dilution of the HF transcriptome content between the two sample methods, which generates either an enrichment or dilution effect on the cell type content. This shift in the abundance of different cell types simply stems from the fact that skin biopsy samples cover interfollicular skin areas (whole tissue sampling) that would either be poorly or not obtained at all in plucked hair samples. Therefore, the restricted plucked hair method of tissue sampling can enrich specific cell populations within and in connection to the HF. In contrast, interfollicular cell populations that are more abundant in the skin and are less connected to HF can be enriched in skin biopsies. Choosing an appropriate sample collection method is essential in such cases when the sought-after cell population is more abundantly present within the HF.

Despite the abundance of studies performed on human and mouse skin tissues reporting new findings and further identifying new populations and subpopulations of cells, current knowledge about skin’s complexity is limited, and further research is needed. This is so even considering that humans are one of the most extensively studied mammals, and that the murine epidermis is one of the best studied mammalian tissues [4]. However, before this study, it was unclear which cell populations consistently remained in plucked hair, i.e., which populations were lost and which were retained with the plucked hair shaft. Due to the lack of an equine-specific study and the availability of a murine-specific scRNA-seq in combination with spatial mapping, a murine skin-specific comprehensive study [5] was used to identify the cell populations obtained with the plucked-hair and skin-biopsy sample-collection methods. Using this data, which identified genes uniquely expressed in different cell types within the mouse skin, we contrasted the abundance of those cell type-specific genes in horse plucked hairs versus skin biopsies. Subsequently, this led to the identification of the different cell types enriched in the two sample sources.

Genomic studies often aim to identify genetic polymorphisms associated with various phenotypes and disorders [6,7,8]. Studying the effect of those polymorphisms on gene expression is a logical step in functional, genetic studies that are limited to the tissues that express these genes and manifest the phenotype. This is specifically important in complex traits that are often associated with regulatory polymorphisms. Collecting specific body tissues are hampered by several factors, such as the accessibility of certain tissues, the invasiveness of the sample-obtaining method, and the associated animal welfare and ethical considerations [9]. For these reasons, this study aims to compare the use of non-invasive hair-plucking method with the invasive skin-biopsy sample collection method using Icelandic horses [10] as a mammalian model. The resulting outcomes might benefit gene expression studies of skin-related traits in horses and other domesticated species. We also aim to investigate which cell types can be enriched or diluted in plucked hairs compared to skin biopsies. With this, we aimed to provide researchers interested in integumentary structures, particularly equine researchers, with information on targeted cell types for each method. Since gene expression studies depend on capturing the affected cell types, we initiated this study to facilitate the choice of the most humane sample collection method.

## 2. Results

### 2.1. Non-Invasive Hair-Plucking and Invasive Skin-Biopsy Sample-Collection Methods Yield Comparable Quality and Quantity of RNA-Seq Results

The performed RNA-seq experiment yielded an average number of 28.7 million reads (range: 19.6–38.2 M) in the different samples. The read counts in the plucked hair samples ranged from 20.0 to 38.2 M and in the skin biopsies from 19.6 to 37.7 M. Mapping the reads to the latest horse reference genome sequence assembly (EquCab3) [11] revealed the expression of 14,749 genes. Among which, 14,170 were protein-coding genes, which is about 67% of all protein-encoding genes annotated in the horse genome assembly (*n* = 21,129). The lowest total number of expressed genes was detected in plucked hair samples from the mane (*n* = 14,547), whereas the highest number of genes was recorded in the case of RNA extracted from the skin biopsies at the forelock area (*n* = 14,749). It should be stressed that only genes with at least one read per million (RPM) and expressed in at least two samples were considered in this study. Similar results in the quality and quantity of the RNA-seq reads indicate that either method can be used in transcriptomic studies without noticeable technical differences, and both yield comparable results.

### 2.2. Differences in Transcriptome between Plucked Hairs and Skin Biopsies Affect Keratin-Related Genes and Cellular Process Enzymes

The difference in the transcriptome between plucked hair samples and skin biopsies was investigated (Figure 1A). Here, 1964 mRNAs were expressed at higher levels, and 1838 were expressed at lower abundance (Table 1). Additionally, keratin-associated genes (KRT84, and KRT75) together with the RHCG gene were found among the most significantly differentially expressed genes in the plucked vs. biopsies contrast.

This result was further split by performing two subsequent comparisons: plucked hair samples vs. skin biopsies from the forelock location (Figure 1B) and another contrast from the mane location (Figure 1C). Although the plucked-hair and skin-biopsy samples were obtained from the same horses, the contrast in the forelock identified 1168 with higher abundance and 1094 with lower abundance genes, while the mane contrast identified 944 with higher abundance and 1091 with lower abundance genes (Table 1). Similar to the previous plucked-hair versus skin-biopsy samples, keratin-encoding genes were found to have a higher abundance in plucked-hair samples. In contrast, several genes encoding enzymes involved in cellular metabolism like MGST1 and MMP7, were expressed at significantly lower levels in the plucked-hair samples (Figure 1B,C).

### 2.3. Validation of the RNA-Seq Results Using qPCR

The RNA-seq results were validated by performing a qPCR experiment using five genes (Figure 2). KRT84 and RHCG mRNAs had higher abundance in the plucked-hair samples in both the RNA-seq and qPCR experiments. Similarly, the higher abundance of FGL2 and MGST1 observed in the RNA-seq experiment in the skin biopsies was validated in the qPCR experiment. The same result was observed for HSPA8, which was not significantly differentially abundant in the RNA-seq and qPCR experiments.

### 2.4. Differential mRNA Abundance Is Due to Different Amounts of HF-Associated and Interfollicular Cell Populations between Plucked Hair and Skin Biopsies

Considering the differences in mRNA abundance, we hypothesize that the observed variability could be related to the differential presence of various cell types in the material derived from plucked hair and skin biopsies. To verify this hypothesis, we used a dataset of single-cell RNA-seq (scRNA-seq) supported with spatial mapping that identified genes differentially expressed in seven main cell populations (i.e., permanent epidermis keratinocyte, skin fibroblasts and fibroblast-like cells, anagen HF keratinocyte, vascular cells, neural crest-derived cells, immune cells, and what the authors called miscellaneous cells). In total, 56 different cell types and states of mouse skin samples were identified during the anagen and telogen phases [5].

The overlap between the pattern of gene expression detected in our equine mRNA-seq and the genes found to have a statistically significant differential abundance in the plucked hair versus skin biopsy contrast is represented along with the genes detected in the murine scRNA-seq experiment (Figure 3). Moreover, the genes statistically significantly expressed in any of the cell populations recognized in the murine spatial scRNA-seq dataset were also identified and represented (Figure 3). An amount of statistically non-significant genes corresponding to 2562 genes detected only in the horse dataset, 6112 genes detected only in the mouse dataset, and 6158 genes common for both datasets were identified and excluded from further analysis. Additionally, 1115 and 460 statistically significant genes expressed in horse or mouse datasets were excluded from further analysis. Although the remaining 4914 genes were expressed in both datasets, 2870 and 1400 genes were excluded from further analysis because they showed a statistically significant difference in either the horse or mouse datasets. Finally, 644 genes found to be expressed and statistically significantly differentially expressed in both datasets were used in the identification process of the cell populations enriched in the plucked hair versus skin biopsy methods.

Subsequently, genes significantly expressed in more than one population were excluded from the list of the 644 genes. The exclusion of such genes resulted in the identification of 42 cell type-specific genes, hereafter called cell markers. Each gene of these cell markers is uniquely expressed in only one of the seven main cell populations. Finally, the abundance of those cell markers was investigated in the plucked hair versus skin biopsies comparisons (Figure 4).

The differential abundance of the cell-type specific gene-marker sets between the plucked hair and skin biopsy samples provided conclusive results about which cell types were more abundant in each plucked hair and skin biopsy samples (Figure 4). It was shown that all cell marker genes expressed in the permanent epidermis keratinocytes (5/5), immune cells (2/2), miscellaneous (red blood and muscle) cells (2/2), and the majority of cell marker genes expressed in skin fibroblasts and fibroblast-like cells (10/13), and vascular cells (5/6) had higher abundance in the skin biopsies as compared to the plucked hair samples. In contrast, the majority (8/9) of the gene marker specific to the anagen HF keratinocytes showed a higher abundance in the plucked hair samples than in skin biopsies. However, in the neural-crest-derived cells, the results were inconclusive, with one-half (3/5) of the genes had a high abundance in skin biopsies and the other half (2/5) had a high abundance in plucked hair samples (Figure 4).

To clarify the results of the neural-crest derived cells, the expression of the five gene markers for this population was further investigated in its Schwann and melanocyte sub-populations. CNP and MPZ were found to be markers of Schwann cell subpopulation while ATP6V1G1, NPY and PMEL were found to be markers of the melanocyte subpopulation. This result indicates that the Schwann cells are enriched in the skin biopsies, whereas the melanocytes are predominantly enriched in the plucked hair samples.

## 3. Discussion

Skin is a complex tissue with several structures and different cell types that inter-talk to maintain skin integrity. This study’s plucked hairs vs. skin biopsies comparison identified 3802 mRNAs with significantly differential abundance related to tissue specificity. The finding that keratin-associated genes were more abundant in plucked hair samples while metabolic enzymes were more abundant in skin biopsies supported this interpretation.

This difference in gene expression could be explained by the fact that one of the skin’s basic structures is HFs, which are organized into several substructures and composed of a number of cell types with distinct gene expression characteristics. There are three phases in the life cycle of a HF [12], beginning with the anagen phase, during which the follicle actively produces a growing hair fiber from the highly proliferative matrix keratinocytes adjacent to the dermal papilla, and the follicle as a whole attains its maximal length. Second, is the brief catagen phase which removes the lower two-thirds of the epithelial component of the follicle and reorganizes the dermal components, while the telogen phase follicle does not produce a hair fiber but is in a state of resting prior to resumption of a new anagen phase [12,13,14].

The size and components of the follicle are thus significantly different at different stages of the hair cycle. Consequently, the cellular content obtained from plucking hair fibers in different hair cycle stages also differs. In humans, plucking of anagen phase follicles yields the most cellular material, sometimes including almost the entire lower follicle, with its most peripheral dermal components, such as keratinocytes of inner and outer root sheaths and the hair matrix. While plucking telogen hairs collects less material, typically yielding the sac of HF keratinocytes ensheathing the proximal hair fiber [15,16].

The relative durations of anagen and telogen phases differ between species and body sites. For example, in adult rodents, most of the hair fibers at any given time are in telogen, while on the human scalp, most hair fibers are in the anagen phase, which lasts for 2–8 years at this site [14,17]. Moreover, most HFs in the horse mane and forelock are in the anagen phase [18], making them a good source of tissue when plucking hair. Compared to cellular material obtained from plucking anagen-phase hairs, sampling using skin punch biopsies is more invasive. Still, it collects interfollicular cell types that are assumed to be missing in plucked hair samples.

This difference in the cellular material obtained using plucked hairs and skin biopsies was studied by investigating the abundance of gene expression detected using the two sample collection methods. In the plucked hair samples, there was an increased abundance of transcripts encoding root sheath (KRT75) and hair (KRT84) keratins and Keratin Associated Proteins (ENSECAG00000010582 and ENSECAG00000040146). These keratins are readily detectable by proteomic analysis of hair fibers of both mouse [19] and human [20]. Whereas the mRNA of many genes involved in cellular processes, such as MMP7, FGL2, and MGST1, were found to have a higher abundance in skin biopsies (Figure 1). We believe that this difference is not a result of a differential gene expression but is rather due to the variability in the proportion of cell types derived from the two sampling methods. We suppose that the hair fiber-associated cell populations are enriched in plucked hair samples but constitute a smaller fraction of cells in bulk skin biopsies.

We anticipate that plucked hair samples are more suitable than skin biopsies for genetic studies of phenotypes such as coloration, structure and growth of equine mane hair. We are currently performing studies that utilize this method to study different aspects of the mane and coat. One reason the plucked hair method is more suitable than skin biopsies in specific studies, such as mane hair color, than skin biopsies is that, sometimes, the sought-after cell type affecting the mentioned phenotypes is diluted in bulk skin biopsies and enriched in plucked hair samples. Another reason is when the sought-after cell type can be equally obtained from either of the two sample collection methods. The latter case occurred in our results regarding the neural crest-derived cells and probably the cell populations beneath it, especially dermal papilla.

The transcriptomes obtained from the non-invasive sampling of hair follicles and the invasive sampling of skin biopsies from the mane and forelock locations of the horse enabled thorough comparison of these two methodologies both qualitatively and quantitatively. Our conclusion is that the non-invasive plucking of hair follicles can be more suitable, depending on the research question, to capture differential gene expression patterns and subsequent functional studies. We also anticipate that the method can be applied not only in humans, but also in other mammalian species [21]. Results obtained using the plucked hair method may be more distinct because the levels of gene expression of individual genes becomes less diluted by the surrounding tissue, as supported by the observation of higher abundance of anagen hair follicle keratinocyte cell markers (Figure 4). In contrast, the skin-biopsy sample-collection method is indispensable for studying interfollicular cell populations and telogen hair follicles.

## 4. Materials and Methods

### 4.1. Animals and Sample Collection

We sampled seven skin biopsies (four from the forelock and three from the mane locations) from four Icelandic horses kept in Swedish farms at the time of sample collection, with consent from their owners. One of the authors (RN), with the help of a veterinarian, collected the hair samples consisting of 30–50 hairs plucked from the same locations of the biopsies. The hairs were grasped at 5–10 cm away from the skin and pulled at a 90° angle with a fast movement. The hairs were then cut at a length of <1 cm away from the root at the time of collection, and placed in cryotubes containing RNA-later (Thermo Fisher Scientific, Waltham, MA, USA).

On the same horses sampled for hair, a veterinarian dissected the skin biopsies after moderately sedating the horse by an intravenous injection of Detomedin (Domosedan vet) (Orion Pharma, Danderyd, Sweden), dosing according to FASS VET. Local anesthesia was performed by subcutaneously injecting the horses with approximately 1–2 mL of Xylocaine (10 mg/mL) with adrenalin (5 µg/m) (Aspen, Aspen, CO, USA) at the place of the biopsy. Once the horse got sedated, skin punches with a diameter of 6 mm (Integra LifeSciences, Princeton, NJ, USA) were used to dissect the skin biopsies from the forelock and middle of the mane (Figure 5). Biopsies were deep enough to include all skin layers down to the subcutaneous fat layer, which was manually removed before the RNA extraction step. One to two cross-stitches using Supramid surgical monofilaments (B. Braun, Melsungen, Germany) were performed to close wounds before they were cleaned. Skin biopsies were placed in cryotubes containing RNA-later (Thermo Fisher Scientific).

Plucked hair samples and skin biopsies were left at RT for about 3 min after collection and then flash frozen in a liquid nitrogen container during transportation. Samples were then stored at −80 °C freezers until RNA extraction was performed. The locations from which the hair and skin biopsy sampels were obtained are illustrated in Figure 5.

### 4.2. RNA Extraction

About 25–30 plucked hairs and about one third of each skin biopsy were taken out from the RNA-later buffer and used for RNA extraction. The samples were first homogenized using Precellys hard-tissue homogenizing tubes with ceramic beads and homogenized on a Precellys Evolution instrument (Bertin Instruments, Montigny-le-Bretonneux, France). After homogenization, the RNeasy kit (Qiagen, Hilden, Germany) was used to extract total RNA. DNase I was used during the process to digest genomic DNA from the total RNA samples. RNA purity and concentration were quantified on a NanoDrop instrument (Thermo Fisher Scientific). RNA quality was measured on a TapeStation instrument using the high sensitivity RNA screenTapes (Agilent, Santa Clara, CA, USA).

### 4.3. Library Prep

The Dynabeads mRNA purification kit (Thermo Fisher Scientific) was used to extract all RNA molecules with a poly-A tail. An amount of 1–2 µg total RNA was used as input. Eluted RNA was used as input using the Corall kit (Lexogen, Vienna, Austria) according to the manufacturer’s instructions and targeting an insert size of 350 bp.

### 4.4. Sequencing

The RNA ready-to-sequence libraries were sequenced on two lanes of an S4 flow cell of a Novaseq6000 (Illumina, San Diego, CA, USA) using v1.5 chemistry reagents. Sequencing was performed at the SciLife SNP&SEQ platform, Uppsala, Sweden.

### 4.5. Analysis

Multiplexed raw fastq files were delivered to us from the sequencing platform. The iDemuxCPP software was used to demultiplex the raw fastq files. Quality control was performed using FastQC, and sequencing adapters were removed using TrimGalore. Mapping was performed using STAR software to the EquCab3 horse reference genome assembly [11], and GTF annotation files, both obtained from ensemble.org. Feature counts was performed using the Subread tool kit [22]. The feature counts were finally analyzed using the EdgeR package [23]. A minimal count of one per million reads in at least two samples was used to filter out all genes with low expression. The trimmed mean of M-values (TMM) method, proposed by Robinson and Oshlack [24], was used to normalize the feature counts.

### 4.6. qPCR Validation

cDNA synthesis was performed using the High Capacity cDNA Reverse Transcription kit (Thermo Fisher Scientific) according to the manufacturer’s protocol. This was followed by performing a qPCR experiment using SYBR Green PCR Master Mix (Thermo Fisher Scientific) and run on a StepOne instrument (Thermo Fisher Scientific) using the primers listed in Table 2. Actin-beta was used as a housekeeping gene.

### 4.7. Cell Type Idéntification

A list of genes expressed in different murine skin cell types (Permanent epidermis keratinocytes, fibroblasts and fibroblast-like cells, anagen hair follicle keratinocytes, vascular cells, neural crest-derived cells, immune cells, and miscellaneous cells—muscle and red blood cells) was previously identified (Joost et al., 2020). Out of this list, the genes detected with significantly differential abundance in the RNA-seq experiment presented in this study were used to identify the cell types present in each of the four sample sources (forelock plucked hair, mane plucked hair, forelock skin biopsies, and mane skin biopsies).

## Figures and Tables

**Figure 1 ijms-24-00561-f001:**
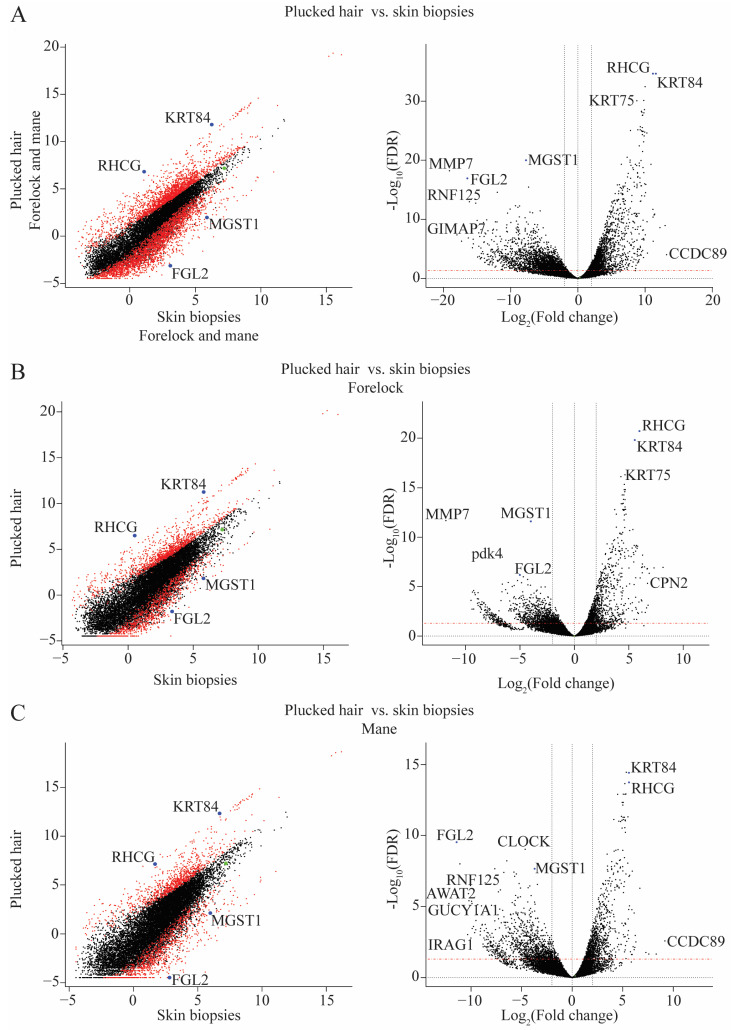
Scatter plots showing the Log2 average expression (**left**) and volcano plots visualizing the difference in the transcriptome (**right**) between plucked hair samples and skin biopsies from the forelock and mane locations combined (*n* = 7 + 7) (**A**), from the forelock location only (*n* = 4 + 4) (**B**) and from the mane location only (*n* = 3 + 3) (**C**). The positive Log2 values in the volcano plots represent genes upregulated in skin biopsy samples, while the negative Log2 values represent genes upregulated in plucked-hair samples.

**Figure 2 ijms-24-00561-f002:**
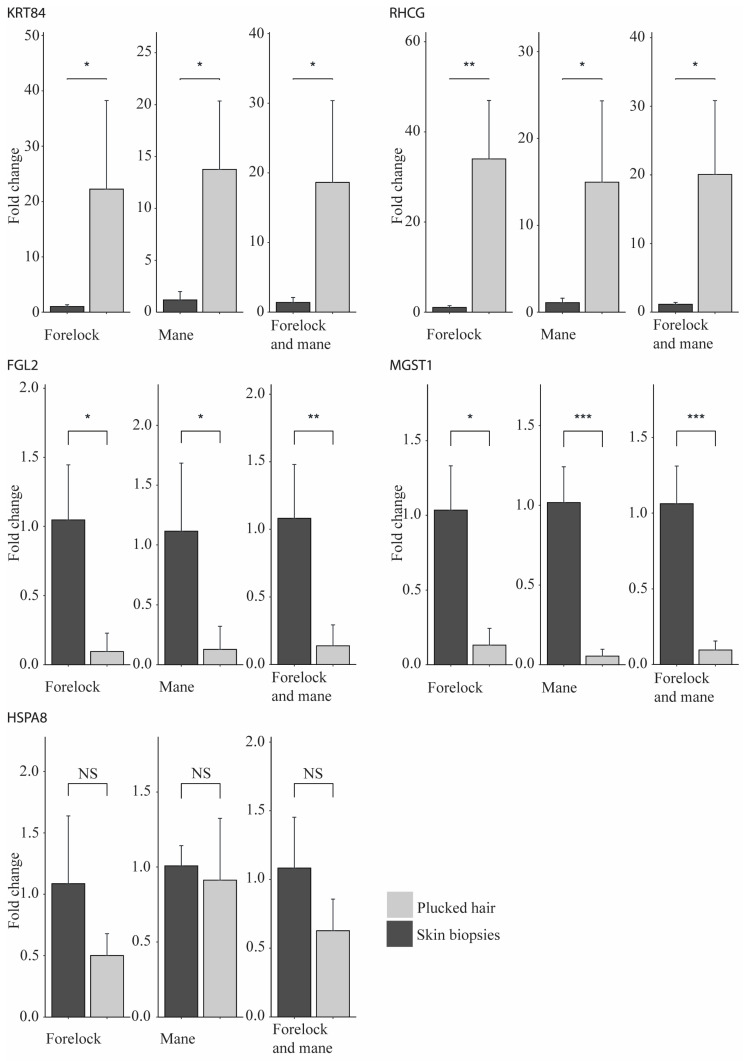
Barplots showing the qPCR results of four upregulated genes in the RNA-seq experiment, plus one control gene. Two-tailed Student’s *t*-test was used (*n* = 7) to calculate *p*-values. NS: Non-significant, * *p* < 0.05, ** *p* < 0.01, *** *p* < 0.001.

**Figure 3 ijms-24-00561-f003:**
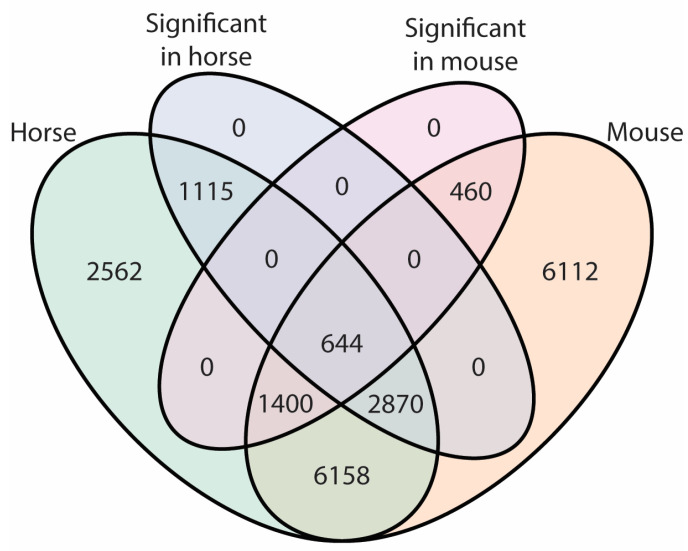
A Venn diagram representing the number of genes expressed in the equine total RNA-seq, murine single-cell RNA-seq, and the amount of statistically significantly differ- initially expressed genes in each of them.

**Figure 4 ijms-24-00561-f004:**
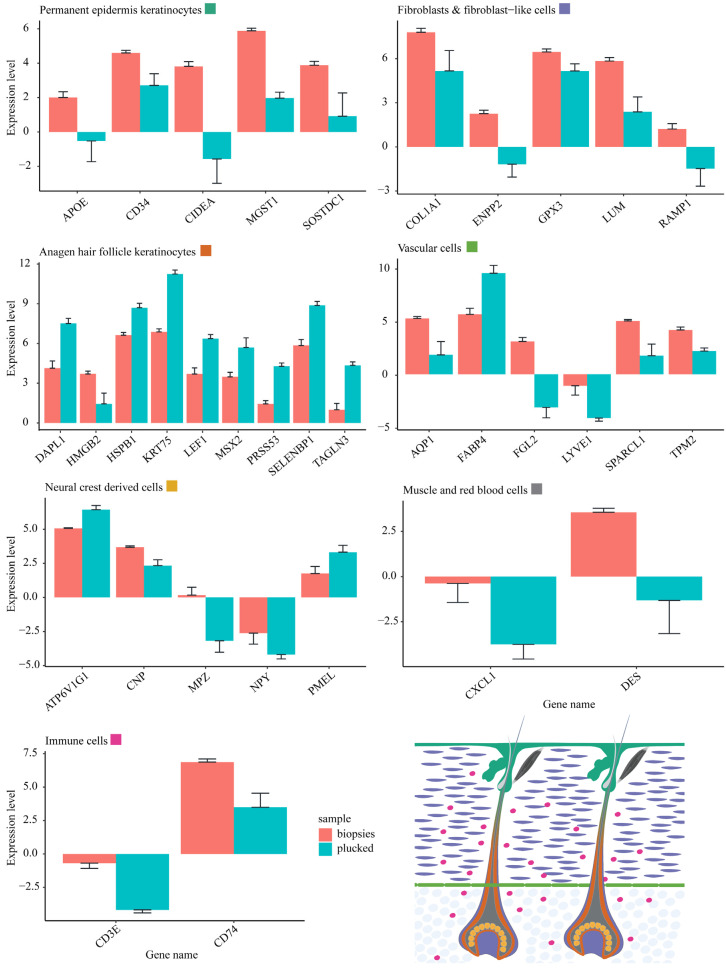
Expression of marker genes for particular populations of cells present in the skin. Only differentially expressed genes between plucked hair and skin biopsies were considered. Each cell population is assigned a distinct color represented in the rectangles adjacent to the cell population name. The spatial localization of these distinct cell populations is illustrated in the longitudinal skin section.

**Figure 5 ijms-24-00561-f005:**
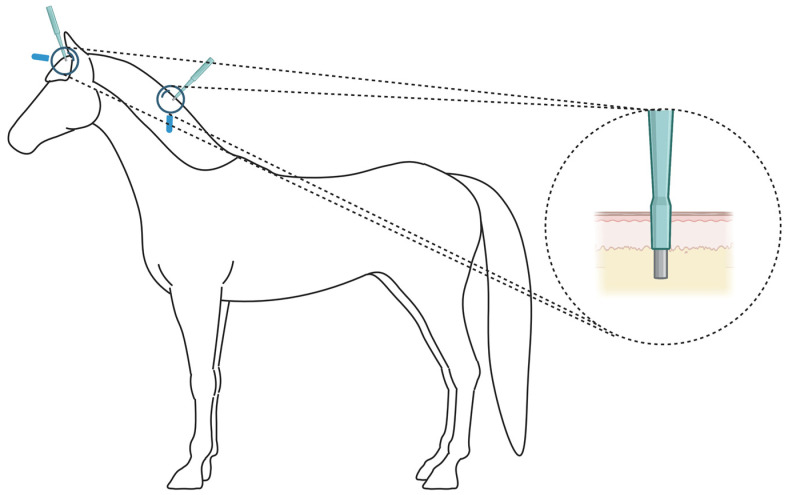
An illustration showing the location of the skin biopsies that were dissected from the forelock (between the ears) and the middle of the mane. The depth of the biopsy that included all the layers above the epidermal fat layer is illustrated. Plucked hair samples were obtained from the same locations. Figure created with BioRender.com.

**Table 1 ijms-24-00561-t001:** The number of DEGs.

	Number of DEGs ^1^	
Comparison	Lower Abundance	Higher Abundance
All Plucked hair versus all skin biopsies	1838	1964
Forelock plucked hair versus forelock biopsies	1094	1168
Mane plucked hair versus mane biopsies	1091	944

^1^ Statistically significant only.

**Table 2 ijms-24-00561-t002:** The sequence of the primers used in the qPCR experiment.

Gene	Forward Primer	Reverse Primer
KRT84	GGCTTTGGCTACAGAGTTGG	TGTCGATGAAGGAGGCAAAT
RHCG	TGATGATCTTCGTGGGCTTC	TGTAGCCTTCCCTGTAGCAGT
MGST1	CCCACCTGAATGACCTTGAA	CAGCCTGTAAGCCATTGACA
FGL2	ACTATTCAGCTCCCGAAGCA	TCTGTGCCAGGTAACAGCAG
HSPA8	ATCTTGGCACCACCTACTCG	GGGTTCATTGCAACTTGGTT
ACTB	CGAGCACGATGAAGATCAAG	GTGGACAATGAGGCCAGAAT

## Data Availability

The Fastq files were deposited at the SRA with accession number PRJNA892645.

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
