# Peer review of "The Enrichment of Specific Hair Follicle-Associated Cell Populations in Plucked Hairs Offers an Opportunity to Study Gene Expression Underlying Hair Traits"

_ijms, 2022, doi:10.3390/ijms24010561_

Round 1
Reviewer 1 Report
Review: The enrichment of specific hair follicle-associated cell populations in plucked hairs offers an opportunity to study gene expression underlying hair traits.
The authors present a well written manuscript describing an alternative method of hair plucking to skin biopsies for evaluation of cell type enrichment. The differences in gene expression profiles between skin and hair follicles are unsurprising and, as the authors point, this has a limited application to a small subset of research question as the hair follicles are unlikely to represent most traits and disease pathogeneses for the skin.
A limitation to this study was that there was no comparison to skin biopsies located further from the forelock and mane as these could have yielded different results as the skin closest to hair growth may have a different population compared to skin at a further distance.
Discussion: Remove the word “Authors” from line 232
Lines 238-255: Is there a difference between the hair follicles that produce the mane and tail versus coat as these are very different structures and could explain some of the differences in gene expression with keratin in the plucked samples. However, if the biopsy samples did not contain coat and only mane/tail (depending on specifically where those samples were taken which is still unclear in the materials and methods) then coat color analysis (one of the mentioned uses) may not be comparable as the coat color is different than mane color and should be discussed.
Reviewer 2 Report
Supplement and clarify the conclusionion.
Reviewer 3 Report
The authors investigated differences in gene expression between non-invasive hair-plucking samples and the invasive skin-biopsy samples. The differences in expressed genes revealed that the types of cells collected in each sample were different. In this study, the authors showed that 1) gene expression analysis is possible from non-invasive hair-plucking samples, and 2) non-invasive hair-plucking samples contain more hair follicle cells. While this study provided interesting information about hair-associated cells, this may be positioned as a pre-analysis for the authors' next study.
The paper is well organized, but it was difficult for this reviewer to understand structures of the abstract and introduction.
Line 16-23: This part should be placed at the end of the abstract. This looks like conclusion.
Lines 41-57: This part should be placed at the end of the introduction.
Lines 106-111: This part is a conclusion rather than an introduction. It should be removed from the introduction or moved to the end of the discussion.
Minor:
Line 232: "Authors Skin" What does this mean?
Line 351: A limited number (Six genes) of qPCR are being performed, is this sufficient?
